# Comparison of Hyperspectral Techniques for Urban Tree Diversity Classification

**Charlotte Brabant** [1], **Emilien Alvarez-Vanhard** [2], **Achour Laribi** [1], **Gwénaël Morin** [2], **Kim Thanh Nguyen** [2], **Alban Thomas** [2] **and Thomas Houet** [1,*]

[1]  CNRS, Université Rennes 2, Unité Mixte de Recherche 6554 LETG, Place du Recteur Henri le Moal, 35043 Rennes CEDEX, France; brabantcharlotte@yahoo.fr (C.B.); achour_eco@yahoo.fr (A.L.)

[2]  CNRS, Université de Rennes 2, Unité Mixte de Recherche 6554 LETG, Place Recteur Henri le Moal, CS 24307-35043 Rennes CEDEX, France; alvarez.emilien@gmail.com (E.A.-V.); gwenaelmorin@laposte.net (G.M.); nkthanh.2812@gmail.com (K.T.N.); alban.thomas@univ-rennes2.fr (A.T.)

*  Correspondence: thomas.houet@univ-rennes2.fr; Tel.: +33-2-99-14-17-20

**Abstract:** This research aims to assess the capabilities of Very High Spatial Resolution (VHSR) hyperspectral satellite data in order to discriminate urban tree diversity. Four dimension reduction methods and two classifiers are tested, using two learning methods and applied with four in situ sample datasets. An airborne HySpex image (408 bands/2 m) was acquired in July 2015 from which prototypal spaceborne hyperspectral images (named HYPXIM) at 4 m and 8 m and a multispectral Sentinel2 image at 10 m have been simulated for the purpose of this study. A comparison is made using these methods and datasets. The influence of dimension reduction methods is assessed on hyperspectral (HySpex and HYPXIM) and Sentinel2 datasets. The influence of conventional classifiers (Support Vector Machine–SVM– and Random Forest –RF–) and learning methods is evaluated on all image datasets (reduced and non-reduced hyperspectral and Sentinel2 datasets). Results show that HYPXIM 4 m and HySpex 2 m reduced by Minimum Noise Fraction (MNF) provide the greatest classification of 14 species using the SVM with an overall accuracy of 78.4% (±1.5) and a kappa index of agreement of 0.7. More generally, the learning methods have a stronger influence than classifiers, or even than dimensional reduction methods, on urban tree diversity classification. Prototypal HYPXIM images appear to present a great compromise (192 spectral bands/4 m resolution) for urban vegetation applications compared to HySpex or Sentinel2 images.

**Keywords:** very high spatial resolution; tree species; training methods; dimension reduction methods; classification methods

## 1. Introduction

Urbanization reflects important environmental transformations, such as a change in the energy balance [1], ecosystem loss or fragmentation, air, soil and water contamination, or even a loss of farming land, as well as an increase in the need for water and a reduction of biodiversity [2–5]. In addition, urban areas are extremely sensitive to global change as they have higher temperatures and atmospheric concentrations of pollutants or fine particles than rural areas [6].

In this context, urban vegetation has several interests: it provides many services such as reducing the urban heat island by providing shade, the evapotranspiration and the photosynthesis of the trees [7–9] and improving air quality or even urban biodiverse habitats [8]. Urban vegetation also presents drawbacks such as the exacerbation of allergies due to pollen, and the provision of habitats for certain bird species considered harmful by the population. However, these services and disservices depend on the plant species, their location and structure [6,10–12]. For instance, their capacity to

mitigate the urban heat island depends on the structure and composition of the species on the one hand, and on their respective water needs on the other hand [13,14]. Regarding these advantages and drawbacks, mapping urban tree diversity is therefore an important issue; herein diversity is defined as the greatest number of species or families that can be distinguished.

Remote sensing applied in urban areas offers the opportunity to overcome a lack of reliable and reproducible vegetation information in private and public lands to more accurately assess ecosystem functions [15–18]. Some authors show the value of VHSR multispectral data such as Pléiades or Ikonos to distinguish tree species [19–21]. For instance, Pu et al. [15] obtained an overall accuracy (OA) of 67.22% with WorldView 2 (eight bands), although the producer's accuracy was not good enough due to this low spectral resolution. Multispectral data such as Sentinel (10–20 m spatial resolution) show also great potential for tree species classification [22,23]. Immitzer et al. [22] obtained an OA of 65% for seven tree species without exploring the full potential of Sentinel2 high repetitiveness. Hyperspectral data also tends to develop for the identification of urban vegetation. While this technology is very spectrally accurate (more than 100 contiguous and narrow bands of about 10 nm each), the spatial resolution of hyperspectral satellite data remains insufficient. Conversely, airborne hyperspectral sensors have demonstrated their capability for tree classification [16–24] (Table 1). Performance is stronger when merged with LiDAR data [17,18,24,25] using three-dimensional information. For example, Alonzo et al. [17] ranked 29 species with an OA of 79.2% using hyperspectral data alone, and 83.4% with additional Lidar data.

**Table 1.** State of this art of VHSR multispectral and hyperspectral images capabilities to discrimination of trees species in urban area.

| Types | Sensors | Spatial Resolution | Spectral Resolution | Overall Accuracies/Number of Discriminated Species | Study |
|---|---|---|---|---|---|
| Multispectral | Pleiades | 0.7 m-Panchromatic 2.4 m-Multispectral | 4 | 63.51% for 7 species | [15] |
| | Ikonos | 1 m-Panchromatic 4 m-Multispectral | 4 | 56.98% for 7 species | [16] |
| | Wordview 2 | 0.46 m-Panchromatic 1.84 m-Multispectral | 8 | 62.93% for 7 species | |
| Hyperspectral | AVIRIS | 3.5 m | 224 | 79.2% for 29 species | [17] |
| | AISA | 2 m | 63 | Summer = 48% for 7 species Fall = 45% | [18] |
| | AVIRIS | 3.5 m | 131 | 69% for 4 evergreen species 70% for 12 deciduous species | [19] |
| | MIVIS | 4 m | 102 | 75% for 7 families | [20] |
| | HySpex | 0.4 m | 80 | 91.7% for 13 species | [21] |
| | HySpex | 2 m | 290 | 69.3% for 5 species | [24] |
| | Rikola | 0.7 m | 64 | 63% for 6 species | [25] |
| | APEX | 2.5 m | 288 | 90% for 7 species | [26] |
| Hyperspectral + LiDAR | AVIRIS + LiDAR | 3.5 m | 224 | 83.4% for 29 species | [17] |
| | AISA + LiDAR | 2 m | 63 | Summer = 57% Fall = 56% | [18] |
| | Hyspex + LiDAR | 2 m | 290 | 83.7% for 5 species | [24] |
| | AISA + LiDAR | 1 m | 366 | 82.12% for 4 species | [27] |

Although the large number of bands of hyperpsectral imagery can contribute to the better discrimination and classification of spectral signatures [28], the induced high dimensionality is one of the main limitations of hyperspectral images. Too numerous spectral bands may provide more confusion than separability, making statistical methods less accurate than with a limited number of spectral bands—this is known as the Hugues phenomenon [29]. This has contributed to the development of dimension reduction methods aimed at selecting the most relevant information. Used to pre-process hyperspectral imageries [30,31], dimension reduction (DR) methods project spectral information into a smaller subspace in order to minimize the amount of data while maximizing the statistical separability of spectral values [32]. Two main DR methods can be distinguished:

- Feature extraction is defined as a set of methods for extracting information from an image [33], such as transformative methods to derive a characteristic [34]. Principal component analysis (PCA) is often used as a DR method because it preserves the greatest spectral variability of original images [35]. However, it does not take account of the signal-to-noise ratio (SNR), unlike the minimal noise fraction (MNF) [36].

- Feature selection methods consist of considering one or more subsets of initial features according to their relevance to the considered question. These methods preserve the interpretability of the original data [37]. In the absence of prior knowledge, a selection is made based on the most informative, least correlated elements in order to preserve the original information as much as possible [38,39]. Vegetation indices are regularly used to provide information on the biogeochemical properties of plants in order to distinguish them. Moreover, these indices allow for better discrimination than when using spectral values from spectral bands alone, for instance in the case of a vegetation class whose composition is too similar [37].

Other factors influence the tree species classification. Beker et al. [40] showed that pixel size strongly influences the accuracy of the classification (for wetland mapping). Further, Dalponte et al. [28] found that the influence of spectral resolution is strongly related to the choice of a classifier. The support vector machine (SVM) and random forest (RF) classifiers are often used for tree species classification with hyperspectral data and are described as the best classifiers for pure pixels [41–44].

The high-dimensional nature of hyperspectral data introduces significant limitations for supervised classification, including the availability of learning samples [45]. Regularly, the number of samples per class is much smaller than the dimensionality of the problem, resulting in weak classification results [46]. One approach is to recognize that individual training samples may affect the classification differently and therefore use the most informative samples [44]. When a limited number of training samples is available, the stratified K-folds cross-validation is often used to split the dataset into training and validation samples, taking into account the proportion of individuals from each class. It is assimilated as one of the most rigorous methods to prevent overfitting and to estimate properly the error rate [47]. But K-folds results depend of the training set and the partition into folds [47]. Alonzo et al. [48] and Clark et al. [38] use stratified 10-fold cross-validation for an unbalanced and limited number of training samples, showing less biased results than with 2-folds while remaining less costly with respect to computation when the number of folds is equal to the number of samples.

This state-of-the-art technology shows that the accuracy of tree species classification using VHRS hyperspectral images can be influenced by DR methods, classifiers, learning methods, defined hereafter as hyperspectral techniques. In this context, evaluating the capabilities of VHSR hyperspectral sensors in order to classify the diversity of tree species in urban area is of prime importance. The aim of this study is to compare and assess which hyperspectral techniques, or combinations, show the greatest performance. Moreover, as spatial and spectral resolutions may influence classification accuracy, another objective consists of assessing which resolution is the most appropriate for urban tree diversity classification. Indeed, most of the listed literature (Table 1) used airborne hyperspectral data with a spatial resolution below 4 m width, which might not be achievable for spaceborne hyperspectral sensors. This study would thus contribute to the design of a future spaceborne hyperspectral sensor (HYPXIM) by assessing the optimal spatial resolutions for urban applications. While limited to tree diversity classification in this study, it has already been assessed for the detection of photovoltaic panels for example [49]. This paper focuses on commonly-used hyperspectral techniques as their respective influence is rarely assessed. While focusing mainly on hyperspectral data and techniques, the comparison with results obtained with Sentinel2 data may emphasize the added-value of the former.

The first part of this article presents the case study of Toulouse, on which this study is focused, the hyperspectral and ground-truth dataset, and all the methods used. The second part describes the results of experiments made to assess the respective and combined influence of hyperspectral techniques. The third part discusses possible perspectives and improvements for tree detection in

urban environments, as well as the potential of spaceborne VHRS hyperspectral imaging for the study of urban areas in general.

## 2. Materials and Methods

### 2.1. Study Area

The studied site is the city center of Toulouse (43°36′16″N, 1°26′39″E), Southwest France (Figure 1). The urban area is strongly artificialized and the city center is characterized by vegetation present both in public areas (along road and river networks such as the nearby Canal du midi and the Grand-Rond roundabout) and in private areas (inner courtyards, private gardens) nestled between relatively dense buildings.

### 2.2. Remote Sensing Data

As shown in the introduction, spaceborne hyperspectral imagery can be very convenient for urban applications. The French National Space Center (CNES) is considering this option by assessing the potential of such a sensor: the HYPXIM hyperspectral sensor is characterized by 192 bands ranging from 0.4 to 2.5 µm with a spatial resolution that may vary from 4 m to 8 m (Table 2). In order to simulate HYPXIM imagery, an airborne hyperspectral dataset was acquired using the HySpex sensor (Table 2) in July 2015. These data have been resampled accordingly to HYPXIM specificities at 4 m and 8 m spatial resolution. Moreover, to highlight the interest of VHSR hyperspectral images, a Sentinel2 multispectral dataset has also been simulated at 10m resolution (for all spectral bands). Atmospheric corrections have been applied using the Cochise (atmospheric COrrection Code for Hyperspectral Images of remote-sensing SEnsors) method [50]. This method assumes a flat ground hypothesis and estimates the water vapor content using a linear regression report (LIRR). It is particularly well suited for hyperspectral imagery [51].

### 2.3. In Situ Samples

Field work was conducted between May 31st and 2nd June 2017. It is assumed that between the data acquisition (2015) and field survey (2017), the tree composition of the city did not change. Indeed, trees are nowadays part of the climate adaptation and mitigation plan of the municipality, declared as protected contributors to cooling urban heat islands and summer heat waves. However, in the case of disease, some removals may occur, but to our knowledge, there were none during this period. This fieldwork led to the identification and location, using a differential GPS (Trimble Geo7x–with an absolute geometric accuracy below 10 cm), of more than 600 trees in the swath of the hyperspectral dataset. This accuracy allows for the extraction of each sample the corresponding spectral signature. Moreover, each inventoried tree—hereafter "individual"—has been classified into types, family, genus and species. A hierarchical nomenclature of 24 families and 43 species was established. This dataset is very unbalanced, e.g., *Platanus hispanica* (London plane) has 144 individuals while other species like *Prunus avium* (Cherry) has only seven individuals.

**Table 2.** Spectral and spatial characteristic of images used.

| Sensor | Spatial Resolution | Bands | Dynamic Range | Spectral Resolution |
|--------|:---:|:---:|:---:|:---:|
| HySpex | 2 m | 408 | | 0.41 to 0.96 µm = 3.64 nm<br>0.96 to 2.5 µm = 6 nm |
| HYPXIM | 4 m | 192 | 0.4–2.5 µm | |
| HYPXIM | 8 m | 192 | | 10.9 nm |
| Sentinel2 (simulated) | 10 m | 4 | 0.4–0.8 µm | 38–145 nm |
| | | 6 | 0.7–2.2 µm | 18–242 nm |
| | | 3 | 0.4–1.3 µm | 26–75 nm |

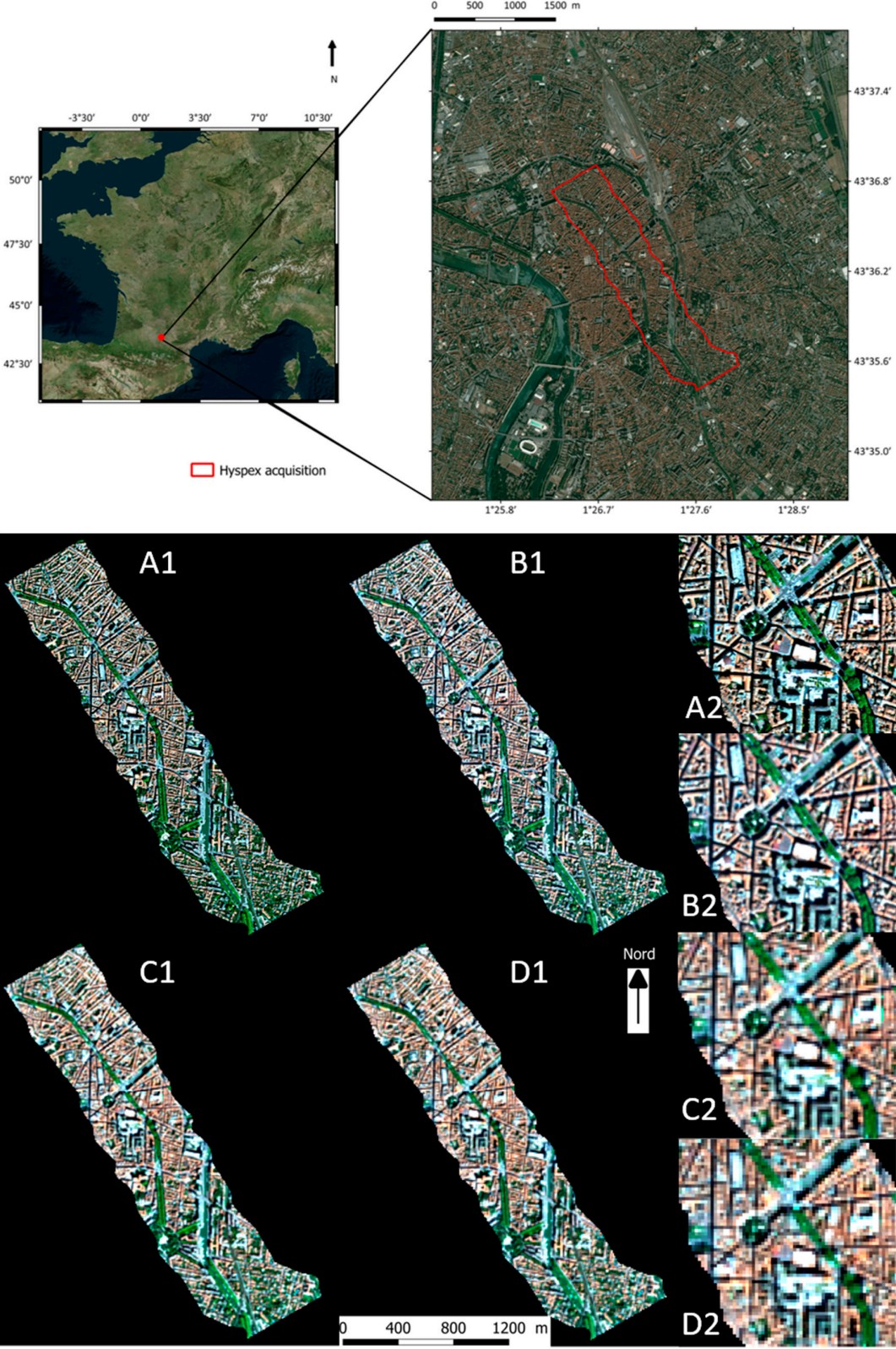

**Figure 1.** True color composite of hyperspectral and Sentinel2 images. (A.1) HySpex image at 2 m; (B.1) HYPXIM at 4 m; (C.1) HYPXIM at 8 m; (D.1) Sentinel2 at 10m. (A.2); (B.2); (C.2); (D.2): Zoom in the city center—RGB (μm): 0.665/0.560/0.490.

*2.4. Methods*

2.4.1. Overall Approach

The aim of this study is to assess the influence of hyperspectral techniques and data on their ability to classify tree diversity. Figure 2 sums up the main hyperspectral techniques used in this study. Several methods of DR (Section 2.4.2.) and classification (Section 2.4.3.) are tested to evaluate which combination of techniques offers the best performance according to the considered hyperspectral images. Each classification performance is assessed using various training size datasets (depending on tree Species and Families nomenclatures, themselves divided into two sub-levels according to the number of available individual samples per class inventoried in the field) and various sizes of stratified k-folds (see Section 2.4.3).

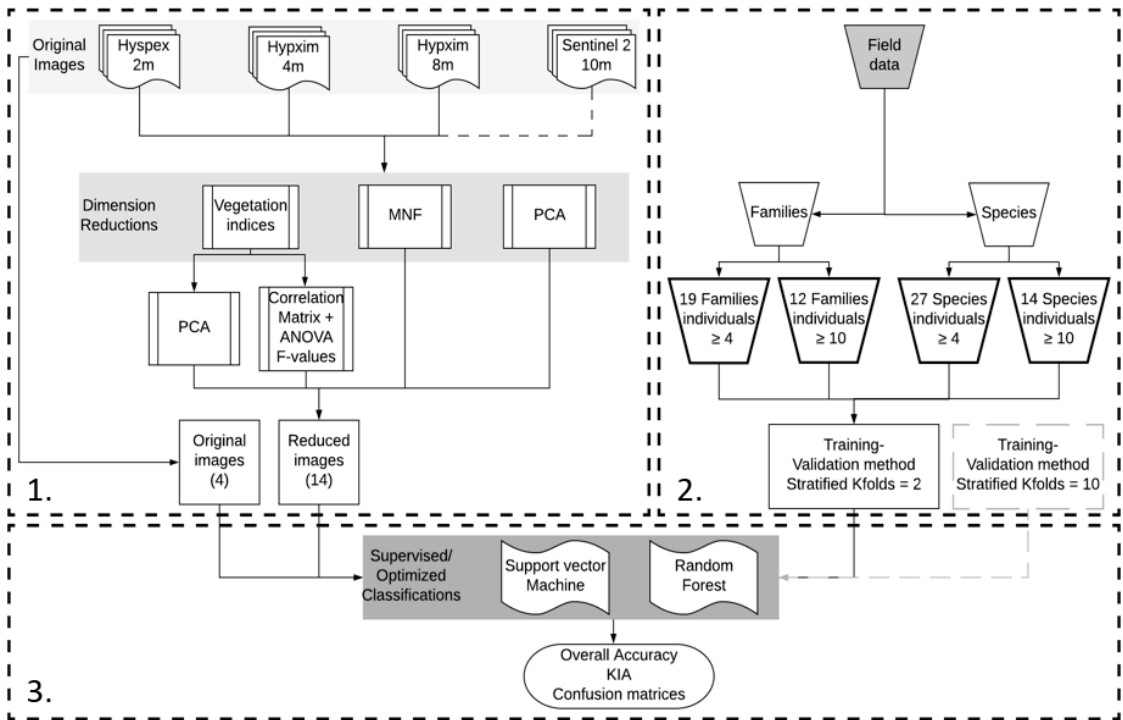

**Figure 2.** General methodological approach. Box 1 lists the tested Dimension Reduction methods providing 14 reduced hyperspectral datasets and compared with the 3 unreduced original ones and the simulated Sentinel2 data—MNF and PCA are also applied on Sentinel2 (cf. Section 2.4.2). Box 2 lists the training data and methods used to train and validate the classification (cf. Section 2.4.3). Box 3 represents the two classifiers and criteria used to evaluate classifications' accuracy (cf. Section 2.4.4).

2.4.2. Dimension Reductions Methods

Feature extraction is defined as a set of methods for extracting information from an image [33], for example, by the use of transformative methods (PCA, MNF). PCA constructs a low-dimensional representation of the data that describes as much of the variance in the data as possible. This is done by finding a linear basis of reduced dimensionality for the data, in which the amount of variance in the data is maximal [34]. However, it does not take account of the signal-to-noise ratio (SNR) unlike the minimal noise fraction (MNF) [37]. The MNF is also a component transformation providing ordered noise-reduced components. Feature extraction methods (PCA, MNF) are applied on the four image datasets (HySpex 2 m, HYPXIM 4 m and 8 m, Sentinel2). The selection of the most appropriate components is made empirically, since they explain more than 98% of variance.

For feature selection methods, the literature brings a lot of information on discriminating indices nevertheless none has been applied in urban contexts. In this study, 96 spectral vegetation indices (VI)

(listed in [37]) have been computed for the three hyperspectral datasets as Sentinel spectral resolution is not sufficient to do so. These VI illustrate various biophysical characteristics such as chlorophyll, leaf area index (LAI), carotenoids, etc., and the selection of the most uncorrelated VI is based on two methods:

- The K-best method uses the F-value of ANOVA (analysis of variance) as a ranking criterion. It compares the interclass and intra-class dispersion between VI. Subsequently, from a correlation matrix (Pearson's r), if the correlation between two VI is greater than 85% (r > 0.85) then the VI with the smallest F-value is removed from the selection.
- A PCA method is used to eliminate the correlation between VI. Kaiser's criterion [39] is used for the selection: components with a eigenvalue greater than 1 represents the same amount of information as a single variable.

Finally, DR methods provide 14 reduced hyperspectral datasets which will be compared with original (unreduced) hyperspectral and Sentinel2 datasets to assess their influence.

### 2.4.3. Training Datasets and Method

Two levels of tree nomenclature are considered: Families and Species. Types have been discarded because of their too general level, the unbalanced dataset but also because multispectral data are known to correctly classify this level of nomenclature. Families and species are respectively sub-divided into two training datasets, depending on the number of individuals per class: the first ones are less restrictive with at least four individuals, while the others consider at least 10 individuals per class. The number of tree classes considered depends on the representativeness of training data (Table 3).

**Table 3.** Training datasets according to families and species nomenclature of trees and their representativeness (number of individuals surveyed during the fieldwork).

**FAMILIES**

| Familiy Name | Number of Individuals | Datasets | |
|---|---|---|---|
| Aceraceae | 4 | | |
| Betulaceae | 4 | | |
| Cupressaceae | 4 | | |
| Juglandaceae | 4 | | |
| Mimosaceae | 5 | | |
| Rosaceae | 8 | | |
| Salicaceae | 8 | "19 families" | |
| Mixed shrub | 14 | | |
| Arecaceae | 16 | | |
| Lawn | 17 | | |
| Magnoliaceae | 21 | | |
| Fagaceae | 22 | | "12 families" |
| Oleaceae | 23 | | |
| Fabaceae | 26 | | |
| Sapindaceae | 29 | | |
| Pinaceae | 32 | | |
| Ulmaceae | 49 | | |
| Tiliaceae | 80 | | |
| Platanaceae | 144 | | |
| Number of tree classes | | 19 | 12 |
| Total of individuals | | 510 | 473 |

number of individuals > 4
number of individuals > 10

**SPECIES**

| Species Name | Number of Individuals | Datasets | |
|---|---|---|---|
| Silver birch | 4 | | |
| Caucasian Wingnut | 4 | | |
| Pine sp | 4 | | |
| Wych elm | 4 | | |
| Caucasian Zelkova | 4 | | |
| Albizia sp | 5 | | |
| Atlas cedar | 5 | | |
| Deodar cedar | 5 | | |
| Cedar sp | 5 | | |
| Poplar sp | 5 | | |
| Western redbud | 6 | | |
| Cherry | 7 | | |
| Holm oak | 8 | "4 species" | |
| Maple sp | 10 | | |
| Oak | 12 | | |
| Fir sp | 13 | | |
| Shrub mix | 14 | | |
| Dwarf plam | 16 | | |
| Southern magnolia | 16 | | |
| Lawn | 17 | | "10 species" |
| White linden | 17 | | |
| Horse chesnut | 21 | | |
| Black locust | 21 | | |
| Common privet | 22 | | |
| Mediterranean hackberry | 35 | | |
| Silver Linden | 63 | | |
| London plane | 144 | | |
| Number of tree classes | | 27 | 14 |
| Total of individuals | | 487 | 421 |

The training method is based on stratified k-folds [44,47]: the original dataset is randomly partitioned into k subsamples where each class is represented with the same proportion of individuals. To assess the influence of the training size, two approaches are considered. The first one uses k-folds = 2 (2-folds): the dataset is divided into 50% training, 50% validation samples and applied for the four training datasets. More precisely, for each class, half of the individuals are used to extract the spectral signature (or assimilated values of reduced datasets) to train the classifier, and the other half is used to validate the resulting classification. In order to optimize the learning capacity of classifiers, each fold is used iteratively as a training and validation sample. The second one is similar to the first one and uses k-folds = 10 (10-folds). It is implemented only for training datasets with at least 10 individuals per class ("12 Families" and "14 Species").

### 2.4.4. Classification Methods

Two commonly used, but not the only classification methods for hyperspectral data are tested, namely the support vector machine (SVM) and random forest (RF). The SVM projects the original feature space into a space with a higher dimensionality to facilitate the discrimination of classes. The classifier is trained using an optimization process associated with a cost function that makes it possible to linearly separate classes to kernel functions (linear, radial basis function—RBF—polynomial, sigmoidal) applied on the original data allowing greater reliability and classification accuracy [41]. However, it can sometimes lead to an over-fitted model with misclassification [41]. The RF classifier is an ensemble of many decision trees. This classifier is also known to be appropriate for hyperspectral data [42,43,52,53] because it tends to perform a selection of characteristics to build the decision tree during the learning process [43]. Its main advantage relies in the ability to stem over-fitting if the number of forests is sufficient. Increasing the predictive power of the model is performed here with an empirical optimization of three hyper-parameters: the number of forests, the maximum number of features, and the splitting criterion (Gini impurity or entropy for the information gain). For both the SVM and RF, hyperparameter optimization has been grid searched thanks to the available training datasets.

### 2.4.5. Experiments: Assessing the Respective and Combined Influence of Hyperspectral Techniques and Data

The respective influence of hyperspectral techniques and data is evaluated based on their ability to classify tree diversity. Classification accuracy is performed using the overall accuracy (OA) and the Kappa index of agreement (KIA). For the 2-folds learning method and data, each SVM and RF classifier is applied 10 times on each reduced and unreduced image dataset in order to estimate the variance of the OA, leading to the computation of 1120 urban tree classifications. Although these results are appropriate to assess the combined influence of the hyperspectral techniques, their respective influence is difficult to evaluate. We assume that averaged OA and KIA (AOA / AKIA) for each technique computed from these classifications can be useful to evaluate their respective contribution. Moreover, the influence of training methods (i.e., the number of k-folds) is assessed differently as the 10-folds is computationally highly time-consuming. It is applied only on the image dataset exhibiting the best classification results, and then compared with the corresponding results obtained with the 2-folds training method.

The combined effect of hyperspectral techniques is analyzed more precisely thanks to the OA and KIA, accounting for both the producer's and user's accuracy.

## 3. Results

### 3.1. Dimension Reductions: Comparison of Extracted Components

The number of selected components from MNF and PCA is respectively of 40 and 30 for all hyperspectral images, and 13 and 9 for Sentinel2. The feature selection method using PCA provides a

selection of 19 vegetation index for HySpex 2 m (explaining 86.16% of the variance), 19 for HYPXIM 4 m (84.5%) and 17 for HYPXIM 8 m (85.89%). When using the ANOVA and the correlation matrix, 45 uncorrelated VI are selected for HySpex 2 m, 39 for HYPXIM 4 m and 38 for HYPXIM 8 m. A recurrence of selected VI is observed between datasets. For example the VI Datt (850) is selected for all image datasets. HYPXIM 4 m exhibit the larger F-values interval (between 38.26 and 1.96) compared to HySpex (from 34.26 and 0.06) and HYPXIM 8 m (27.96 and 1.73). The larger is the interval (i.e., the variance between selected VI), the greater is the separability power. The most selected VI regardless to the resolution rely to the chlorophyll content with 24/45 selected indices for HySpex, 19/39 for HYPXIM 4 m and 24/38 for HYPXIM 8 m. In addition, the first 10 VI exhibiting the highest F-values are also related to chlorophyll (except "CaCol_515_550" for the 8 m HYPXIM image that is linked to the carotenoid) meaning that this biophysical property has the most discriminating potential (cf. Supplementary Figure S1 for details).

## 3.2. Evaluation of the Respective Influence of Hyperspectral Techniques and Data

When using the 2-folds training method, both classifiers offer similar results. Classifications made with RF presents an AOA = 45.15% and an AOA = 42.61% with SVM (Table 4). The latter has the lower AOA while exhibiting the highest score (AOA = 59% and best AOA = 58.54% for RF). RF shows the greatest variance with on average σ = 4.2% against σ = 3.8% for SVM.

**Table 4.** Urban tree diversity classification averaged overall accuracies (AOA) and averaged Kappa index of agreement (AKIA) obtained using the 2-folds learning method and applied on each reduced hyperspectral image, and on the original (hyperspectral and Sentinel2) image datasets with RF and SVM according to the four sub-levels for families and species datasets.

| | | 19 Families (n indiv. > 4) | | | 12 Families (n indiv. > 10) | | | 27 Species (n indiv. > 4) | | | 14 Species (n indiv. > 10) | | | Mean Values | | |
|---|---|---|---|---|---|---|---|---|---|---|---|---|---|---|---|---|
| | | **AOA** | **σ** | **AKIA** | **AOA** | **σ** | **AKIA** | **AOA** | **σ** | **AKIA** | **AOA** | **σ** | **AKIA** | **AOA** | **σ** | **AKIA** |
| **SVM** | Original images | 36.40%±3.6 | | 0.18 | 39.18%±4.1 | | 0.22 | 35.55%±3.5 | | 0.16 | 41.53%±3.0 | | 0.19 | 38.16%±3.6 | | 0.19 |
| | PCA | 36.44%±4.3 | | 0.18 | 38.86%±3.9 | | 0.20 | 34.54%±3.5 | | 0.14 | 40.04%±3.3 | | 0.17 | 37.47%±3.7 | | 0.17 |
| | MNF | 54.17%±6.0 | | 0.46 | 59.00%±5.9 | | 0.51 | 51.76%±4.9 | | 0.44 | 58.72%±4.3 | | 0.48 | 55.91%±5.3 | | 0.47 |
| | VI ANOVA | 40.92%±3.3 | | 0.28 | 45.15%±3.0 | | 0.33 | 37.63%±3.7 | | 0.23 | 44.37%±4.3 | | 0.29 | 42.02%±3.6 | | 0.28 |
| | VI by PCA | 37.90%±2.0 | | 0.24 | 41.34%±2.2 | | 0.27 | 36.14%±4.7 | | 0.20 | 42.54%±3.0 | | 0.27 | 39.48%±3.0 | | 0.24 |
| | Mean values | 41.17%±3.8 | | 0.27 | 44.70%±3.8 | | 0.30 | 39.12%±4.0 | | 0.23 | 45.44%±3.6 | | 0.28 | 42.61%±3.8 | | 0.27 |
| | | **AOA** | **σ** | **AKIA** | **AOA** | **σ** | **AKIA** | **AOA** | **σ** | **AKIA** | **AOA** | **σ** | **AKIA** | **AOA** | **σ** | **AKIA** |
| **RF** | Original images | 37.64%±3.8 | | 0.25 | 40.99%±3.8 | | 0.28 | 35.76%±4.0 | | 0.23 | 41.74%±4.8 | | 0.26 | 39.03%±4.1 | | 0.25 |
| | PCA | 37.21%±4.6 | | 0.22 | 39.91%±4.8 | | 0.24 | 36.85%±3.3 | | 0.20 | 44.19%±4.5 | | 0.25 | 39.54%±4.3 | | 0.23 |
| | MNF | 53.02%±5.8 | | 0.44 | 57.43%±5.9 | | 0.48 | 51.36%±4.6 | | 0.42 | 58.54%±4.4 | | 0.48 | 55.09%±5.2 | | 0.45 |
| | VI ANOVA | 48.30%±3.7 | | 0.39 | 52.21%±3.8 | | 0.42 | 45.08%±3.5 | | 0.36 | 53.01%±2.8 | | 0.43 | 49.65%±3.5 | | 0.40 |
| | VI by PCA | 40.35%±3.3 | | 0.29 | 42.13%±3.3 | | 0.30 | 40.98%±5.5 | | 0.30 | 46.37%±4.2 | | 0.34 | 42.46%±4.1 | | 0.31 |
| | Mean values | 43.30%±4.2 | | 0.32 | 46.53%±4.3 | | 0.34 | 42.00%±4.2 | | 0.30 | 48.77%±4.1 | | 0.35 | 45.15%±4.2 | | 0.33 |

Legend (AOA): 50–60%; 60–70%; >70%.

For the two considered classifiers, results obtained from the MNF image (with AOA = 55.09% and 55.91% for RF and SVM respectively—Table 4) datasets are always greater than those obtained from other reduced/unreduced image datasets. VI by ANOVA has an AOA of 45.83% (AOA = 49.65% and 49.65% for RF and SVM respectively) and VI by PCA 40.97% (AOA = 39.48% and 42.46% for RF and SVM respectively). PCA and unreduced image datasets show similar AOA (38.50% and 38.60% respectively).

Sub-level training datasets show that those with at least 10 samples (12 Families and 14 Species) help to improve tree classification accuracy compared to those with a minimal number of 4 individual samples. The sub-level "14 species" gives the best results with an AOA = 47.10% against 45.62% for the sub-level "12 families". The sub-level "19 families" (n individuals > 4) produces results slightly higher than the sublevel "27 species" with respectively AOA = 42.23% and 40.56%. The comparison of the 2-folds and 10-folds method as the training method shows the potential of 10-folds method because it significantly improves AOA. When comparing results obtained with these two training methods

applied on MNF image datasets, the 10-folds provides an AOA = 68.26% although AOA = 58.42% with the 2-folds one (Tables 5 and 6).

**Table 5.** Urban tree diversity classification results obtained using the 10-folds learning method and applied on reduced MNF image datasets, with SVM and RF classifiers according to subsets exhibiting at least 10 individuals per class.

| | | 12 Families (n>10) | | | | | | 14 Species (n>10) | | | | | | Mean Values | | | |
| | | SVM | | | RF | | | SVM | | | RF | | | | | | |
| | Spatial Resolution | OA | σ | KIA | OA | σ | KIA | OA | σ | KIA | OA | σ | KIA | OA | σ | KIA | |
|---|---|---|---|---|---|---|---|---|---|---|---|---|---|---|---|---|---|
| HySpex | 2 m | 73.81% | ± 7.5 | 0.68 | 73.81% | ± 6.0 | 0.68 | 78.38% | ± 2.5 | 0.73 | 75.68% | ± 2.5 | 0.70 | 75.42% | ± 4.6 | 0.70 | |
| HYPXIM | 4 m | 73.81% | ± 5.0 | 0.68 | 73.81% | ± 6.5 | 0.67 | 78.38% | ± 1.5 | 0.73 | 72.97% | ± 0.5 | 0.65 | 74.74% | ± 3.4 | 0.68 | AOA |
| | 8 m | 59.52% | ± 4.0 | 0.49 | 61.90% | ± 3.0 | 0.51 | 62.16% | ± 3.0 | 0.51 | 64.86% | ± 3.5 | 0.55 | 62.11% | ± 3.4 | 0.52 | 50–60% |
| Sentinel2 | 10 m | 61.90% | ± 5.5 | 0.53 | 59.52% | ± 4.5 | 0.49 | 59.46% | ± 5.5 | 0.48 | 62.16% | ± 3.0 | 0.52 | 60.76% | ± 4.6 | 0.51 | 60–70% |
| Mean values | | 67.26% | ± 5.5 | 0.60 | 67.26% | ± 5.0 | 0.59 | 69.60% | ± 3.1 | 0.61 | 68.92% | ± 2.4 | 0.61 | 68.26% | ± 4.0 | 0.60 | > 70% |

Finally, according to the spatial and spectral resolution, HySpex 2 m show an AOA = 52.60% that is greater than any others image datasets: AOA = 42.38% for HYPXIM 4 m, AOA = 40.51% for HYPXIM 8 m and AOA = 41.93% for Sentinel2 10m (Table 7).

**Table 6.** Urban tree diversity classification results obtained using the 2-folds learning method and applied on reduced MNF image datasets, with SVM and RF classifiers according to subsets exhibiting at least 10 individuals per class.

| | | 12 Families (n>10) | | | | | | 14 Species (n>10) | | | | | | Mean Values | | | |
| | | SVM | | | RF | | | SVM | | | RF | | | | | | |
| | Spatial Resolution | OA | σ | KIA | OA | σ | KIA | OA | σ | KIA | OA | σ | KIA | OA | σ | KIA | |
|---|---|---|---|---|---|---|---|---|---|---|---|---|---|---|---|---|---|
| HySpex | 2 m | 73.81% | ± 7.5 | 0.68 | 73.81% | ± 6.0 | 0.68 | 78.38% | ± 2.5 | 0.73 | 75.68% | ± 2.5 | 0.70 | 75.42% | ± 4.6 | 0.70 | |
| HYPXIM | 4 m | 73.81% | ± 5.0 | 0.68 | 73.81% | ± 6.5 | 0.67 | 78.38% | ± 1.5 | 0.73 | 72.97% | ± 0.5 | 0.65 | 74.74% | ± 3.4 | 0.68 | AOA |
| | 8 m | 59.52% | ± 4.0 | 0.49 | 61.90% | ± 3.0 | 0.51 | 62.16% | ± 3.0 | 0.51 | 64.86% | ± 3.5 | 0.55 | 62.11% | ± 3.4 | 0.52 | 50–60% |
| Sentinel2 | 10 m | 61.90% | ± 5.5 | 0.53 | 59.52% | ± 4.5 | 0.49 | 59.46% | ± 5.5 | 0.48 | 62.16% | ± 3.0 | 0.52 | 60.76% | ± 4.6 | 0.51 | 60–70% |
| Mean values | | 67.26% | ± 5.5 | 0.60 | 67.26% | ± 5.0 | 0.59 | 69.60% | ± 3.1 | 0.61 | 68.92% | ± 2.4 | 0.61 | 68.26% | ± 4.0 | 0.60 | > 70% |

**Table 7.** Classifications mean results by spatial resolution and dimension reductions using the 2-folds learning method.

| | | HySpex (2m) | | | HYPXIM (4m) | | | HYPXIM (8m) | | | Sentinel2 (10) | | | Mean Values | | | |
| | | AOA | σ | AKIA | AOA | σ | AKIA | AOA | σ | AKIA | AOA | σ | AKIA | AOA | σ | AKIA | |
|---|---|---|---|---|---|---|---|---|---|---|---|---|---|---|---|---|---|
| | Original images | 51.67% | ± 4.1 | 0.42 | 30.29% | ± 3.4 | 0.04 | 30.94% | ± 2.6 | 0.03 | 39.76% | ± 4.1 | 0.27 | 38.16% | ± 3.6 | 0.19 | |
| | PCA | 49.95% | ± 4.5 | 0.39 | 32.74% | ± 4.0 | 0.12 | 31.63% | ± 2.4 | 0.04 | 38.05% | ± 4.5 | 0.25 | 38.09% | ± 3.8 | 0.20 | |
| SVM | MNF | 60.79% | ± 6.9 | 0.53 | 59.29% | ± 5.0 | 0.52 | 56.85% | ± 4.4 | 0.49 | 47.96% | ± 4.1 | 0.37 | 56.22% | ± 5.1 | 0.48 | |
| | VI ANOVA | 46.62% | ± 3.8 | 0.36 | 45.00% | ± 3.8 | 0.34 | 34.44% | ± 3.3 | 0.16 | | | | 42.02% | ± 3.6 | 0.28 | |
| | VI by PCA | 38.60% | ± 2.3 | 0.21 | 42.10% | ± 3.5 | 0.31 | 40.20% | ± 3.8 | 0.28 | | | | 40.30% | ± 3.2 | 0.27 | |
| | Original images | 44.71% | ± 3.3 | 0.33 | 41.13% | ± 4.9 | 0.28 | 35.97% | ± 4.1 | 0.20 | 34.31% | ± 4.0 | 0.21 | 39.03% | ± 4.1 | 0.25 | |
| | PCA | 48.19% | ± 5.3 | 0.36 | 32.74% | ± 4.0 | 0.12 | 33.66% | ± 3.3 | 0.11 | 43.57% | ± 4.6 | 0.33 | 39.54% | ± 4.3 | 0.23 | |
| RF | MNF | 60.30% | ± 6.8 | 0.52 | 58.08% | ± 4.9 | 0.49 | 54.02% | ± 4.9 | 0.43 | 47.96% | ± 4.1 | 0.37 | 55.09% | ± 5.2 | 0.45 | |
| | VI ANOVA | 54.56% | ± 4.5 | 0.46 | 49.26% | ± 3.0 | 0.39 | 45.13% | ± 2.9 | 0.34 | | | | 49.65% | ± 3.5 | 0.40 | AOA |
| | VI by PCA | 41.54% | ± 5.3 | 0.29 | 44.61% | ± 3.3 | 0.34 | 41.21% | ± 3.8 | 0.29 | | | | 42.46% | ± 4.1 | 0.31 | 50–60% |
| Mean values (without VI) | | 52.60% | ± 5.1 | 0.42 | 42.38% | ± 4.4 | 0.26 | 40.51% | ± 3.6 | 0.22 | 41.93% | ± 4.3 | 0.30 | 44.31% | ± 4.1 | 0.30 | 60–70% |
| Mean values | | 49.69% | ± 4.7 | 0.39 | 43.52% | ± 4.0 | 0.29 | 40.40% | ± 3.5 | 0.24 | | | | 44.06% | ± 4.0 | 0.31 | > 70% |

### 3.3. Influence of Combined Techniques According to Images

The best tree diversity classification accuracy (OA = 78.38%) is obtained using the reduced image HYPXIM 4 m by MNF, with the sub-level training sample "14 species" (n individuals > 10), the 10-folds training method and with SVM classifier. A similar score is obtained with HySpex 2 m while applying the same hyperspectral techniques and training data and method. KIA are also similar for the both reduced image (KIA = 0.73) but reduced HySpex present a greater variance than reduced HYPXIM at 4 m (σ = 2.5% and σ = 1.5% respectively) (Table 5 and Figure 3). The "12 families" nomenclature produces also good results with OA = 73.81% and KIA = 0.68 with reduced images HySpex and

HYPXIM at 4 m however the latter presents a smaller standard deviation (±5%). HYPXIM 8 m and Sentinel2 show a similar OA = 61.90% (Table 5 and Figure 4).

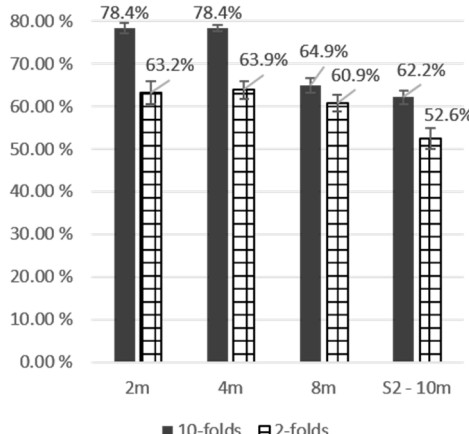

**Figure 3.** Comparison of results (OA) of 2-folds and 10-folds methods for the "species" nomenclature with the MNF DR.

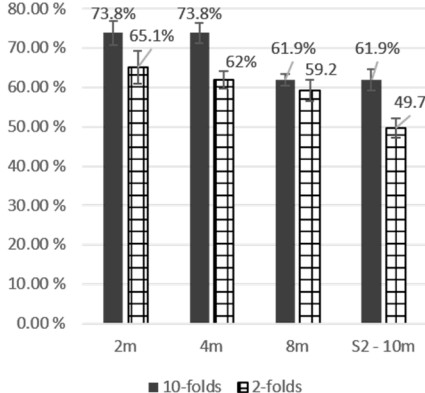

**Figure 4.** Comparison of results (OA) of 2-fold and 10-folds methods for the "families" nomenclature with the MNF DR.

Although the 10-folds always provides higher results compared to 2-folds learning method (with MNF DR only), it reinforces the interest of HYPXIM imageries. Indeed, with the 10-folds learning method, a significant improvement of urban tree diversity classification accuracy is observed: HySpex 2 m and HYPXIM 4 m reduced with MNF both exhibit an OA of 78.38% and 73.81% for 12 families and 14 species classes, respectively (Figures 3 and 4). In the meanwhile, improvements for HYPXIM 8 m and Sentinel2 due to 10-folds are much lower, or exhibit an OA below 65%.

Focusing on the classification images exhibiting the highest OA (Figures 5 and 6), helps to understand classification errors in the context of the obtained scores. Indeed, for the reduced HySpex image, an overestimation of London plane is visible with 2-folds method (Figure 5.1.a), as well as for *Linden sp.* with 10-folds method (Figure 5.2.a). In both cases, London plane is correctly detected on large boulevard (Bd. Lazare Carnot) while over-estimated due to confusion with the shadow of buildings. In specific areas exhibiting a great tree diversity such as the public park of the "Grand-Rond" roundabout, the 10-folds method (Figure 5.2.b) gives a better prediction than 2-folds (Figure 5.1.b) in particular for Silver linden that are correctly detected. Conversely, it contributes to misclassify neighboring pixels of tree alignment on the previously cited boulevard north to the Grand Rond (Figure 5.2.b): Linden are over estimated although only London plane species is present. These misclassifications are less present when classification is applied with reduced HYPXIM image at 4 m (Figure 6). Species are better detected with less errors of neighboring pixels (Figure 6(1.b,2.b)).

However, the classification with 2-folds method (Figure 6.1.b) evinces a bigger misclassification than 10-folds method as for example on the Grand-Rond roundabout where Silver linden are over predicted.

　　Related confusion matrices (Supplementary Table S2) and producer's and user's accuracies (respectively PA and UA – Supplementary Table S3) highlight the influence of the unbalanced in situ dataset. Table 8 illustrates the PA and UA for the classifications shown in Figures 5 and 6.

　　First, the greater the number of individuals per tree species, the better the PA and UA. Then, UA are in most of the cases greater than the PA, except for the well represented tree species. Table S2 shows that tree species with a low PA are in most of the cases mistakenly classified as *London plane*. Third, the number of k-folds significantly improves accuracies for under-represented tree species, particularly for MNF-reduced HySpex images. Finally, the number of missed tree species (0% of PA or UA) is slightly greater with RF than SVM for both 14 species' and 12 families' nomenclatures.

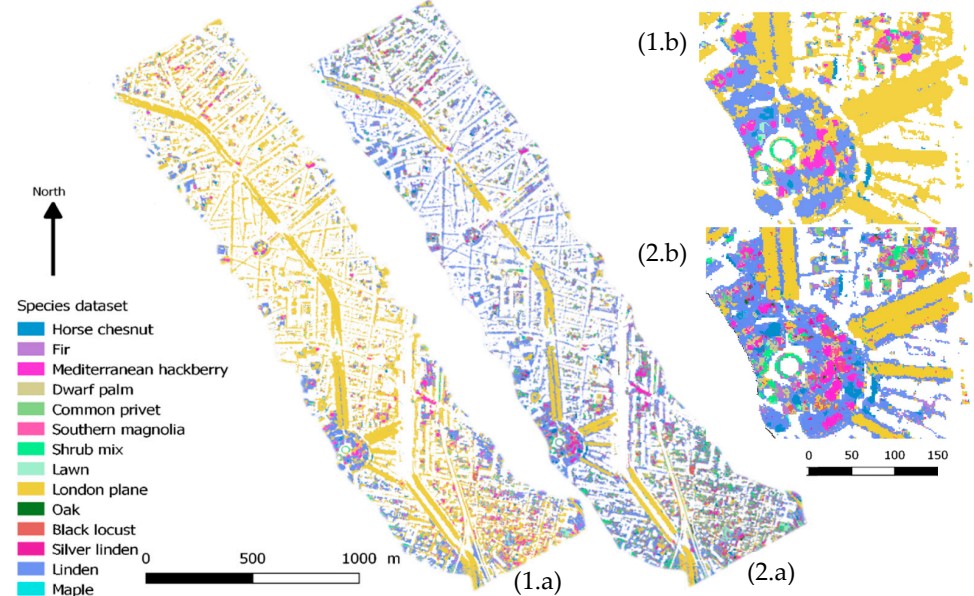

**Figure 5.** SVM classification of trees species ("14 Species" dataset) with MNF dimension reduction applied on HySpex at 2 m. (**1.a**) 2-folds method and (**1.b**) Zoom on Grand-Rond roundabout; (**2.a**) 10-folds method and (**2.b**) Zoom on Grand-Rond roundabout.

**Table 8.** PA and UA for the MNF-reduced HySpex and HYPXIM 4 m hyperspectral images, classified with SVM and trained with the 14 species training dataset with 2-folds and 10-folds. These PA and UA correspond to those obtained for classifications shown in Figures 5 and 6, exhibiting the greatest OA.

| | | | 2-Folds | | | | 10-Folds | | | |
|---|---|---|---|---|---|---|---|---|---|---|
| | | Number of | HySpex 2 m | | HYPXIM 4 m | | HySpex 2 m | | HYPXIM 4 m | |
| | | Individuals | PA | UA | PA | UA | PA | UA | PA | UA |
| **SVM** | Maple | 10 | 0.0% | 0.0% | 0.0% | 0.0% | 40.0% | 66.7% | 20.0% | 50.0% |
| | Oak | 12 | 16.7% | 33.3% | 41.7% | 71.4% | 58.3% | 87.5% | 41.7% | 62.5% |
| | Fir | 13 | 7.7% | 25.0% | 0.0% | 0.0% | 38.5% | 35.7% | 23.1% | 20.0% |
| | Shrub mix | 14 | 0.0% | 0.0% | 28.6% | 44.4% | 21.4% | 27.3% | 35.7% | 55.6% |
| | Dwarf palm | 16 | 6.3% | 25.0% | 50.0% | 57.1% | 25.0% | 26.7% | 37.5% | 66.7% |
| | Southern magnolia | 16 | 25.0% | 33.3% | 62.5% | 55.6% | 56.3% | 52.9% | 50.0% | 53.3% |
| | Lawn | 17 | 47.1% | 80.0% | 52.9% | 69.2% | 58.8% | 71.4% | 58.8% | 62.5% |
| | Silver linden | 17 | 0.0% | 0.0% | 0.0% | 0.0% | 23.5% | 40.0% | 5.9% | 50.0% |
| | Horse chesnut | 21 | 76.2% | 94.1% | 85.7% | 100.0% | 90.5% | 93.5% | 85.7% | 94.7% |
| | Black locust | 21 | 33.3% | 58.3% | 47.6% | 45.5% | 42.9% | 42.9% | 38.1% | 61.5% |
| | Common privet | 22 | 54.5% | 46.2% | 63.6% | 50.0% | 59.1% | 61.9% | 59.1% | 52.0% |
| | Mediterranean hackberry | 35 | 65.7% | 46.9% | 80.0% | 66.7% | 65.7% | 62.2% | 71.4% | 62.5% |
| | Linden | 63 | 87.3% | 49.1% | 55.6% | 43.8% | 69.8% | 57.9% | 74.6% | 46.5% |
| | London plane | 144 | 95.1% | 83.5% | 88.9% | 84.8% | 92.4% | 88.7% | 89.6% | 89.0% |
| | **OA** | | 63.2% | | 63.9% | | 78.4% | | 78.4% | |

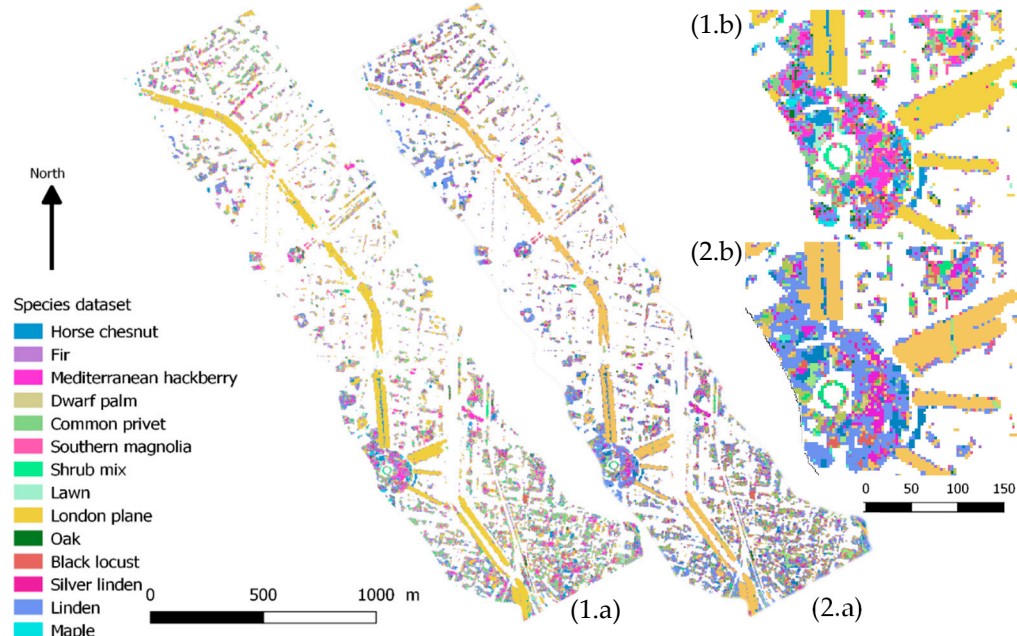

**Figure 6.** SVM classification of trees species ("14 Species" dataset) with MNF dimension reduction applied on HYPXIM at 4 m. (**1.a**) 2-folds method and (**1.b**) Zoom on Grand-Rond roundabout; (**2.a**) 10-folds method and (**2.b**) Zoom on Grand-Rond roundabout.

## 4. Discussion

### 4.1. Evaluating the Respective Influence of Hyperspectral Techniques

First of all, comparing the respective influence of the hyperspectral techniques considered remains a difficult task since the influence of one technique on another is not linear. Although the use of averaged OA values per technique can be criticized, this provides interesting information on the respective contribution of the methods that have been tested for the classification of tree diversity. The method that allows the greatest gain in overall accuracy depends on the learning process and the data. In our case, the gain can be greater than 15% for some image datasets (HySpex 2 m, HYPXIM 4 m) as soon as the number of k-folds with a minimum of 10 individuals per class increases. Second, dimensional reduction methods also improve the classification's overall accuracy than other classification methods. However, depending on the considered image dataset or even the DR methods for instance, some may be punctually more effective than others, fuelling the debate on the justification for choosing one or the other method. Nevertheless, it can be assumed that the very high spatial resolution of hyperspectral data has an inherent limitation: it increases the dimensionality of the data and makes any classification method ineffective if dimensional reduction methods are not previously applied. Indeed, in some few cases, the Sentinel2 image gives similar or even slightly better results than hyperspectral images even though the overall accuracy remains quite low.

While we have focused on dimensional reduction methods, our results are in line with the scientific literature. The most effective method is MNF. The selection of vegetation indices gives interesting results and better averages than the use of unreduced hyperspectral datasets. The MNF's ability to reduce noise while minimizing signal loss explains this level of performance unlike the ACP. Feature selection methods remain less accurate than feature extraction ones. While considering the ANOVA and correlation matrix selection method, low results can be explained by empirically choosing the correlation threshold. Oldeland et al. [54] used a threshold of r = 0.75 without any explanations. Another selection method could be tested such as the Kruskal Wallis algorithm which iteratively tests the best combination of uncorrelated indices [37]. Moreover, these authors have shown that not only vegetation indices can be used, but also various derivatives (1st order or 2nd order) of (part of) the

spectral signature. Indeed, derivatives are insensible to variations in illumination intensity caused by changes in sun angle [55,56] and which may be affected by the floristic composition and vegetation type [37].

### 4.2. Characterizing Urban Tree Diversity with VHRS Hyperspectral Remote Sensing: Advantages, Limitations and Prospects

This study presents results similar to those described in the literature cited in the introduction. Thanks to optimized hyperspectral data and techniques, about 80% of urban tree diversity (14 different species) can be accurately discriminated against. These are fairly good results given that hyperspectral data were acquired in summer, which might not be the optimal season to distinguish species in comparison to late spring or fall.

The spatial and spectral resolutions of HYPXIM images present great potential for such an application. Indeed, in general, the results decrease with the decrease in spatial and spectral resolutions. In some cases, Sentinel2 images give better results than those obtained with HYPXIM. But as soon as hyperspectral techniques are optimized, there is a real qualitative leap allowing HYPXIM 4 m data to obtain results similar to those obtained with HySpex 2 m data. This suggests that 4 m can be an almost optimal spatial resolution for classifying urban tree diversity. On the one hand, at 2 m, even if each tree can be better delineated, classification errors may increase due to the variability of spectral signatures in a crown [48]. Indeed, spectral reflectance can vary within the same tree, depending on biochemistry and water content [57,58], canopy architecture, woody biomass exposure and underlying substrate exposure [30]. Figures 5 and 6 show this effect by comparing classifications from HySpex 2 m and HYPXIM 4 m. North of the 'Grand-Rond' roundabout, the boulevard is entirely composed of *London plane* while between two rows of trees some pixels are assimilated to *Linden* trees with HySpex images. These errors are rare with HYPXIM images at 4 m. On the other hand, a coarser spatial resolution does not help because of mixed pixels [40]. However, spectral resolution also influences the ability to distinguish urban tree species with similar spectral signatures [57]: the choice of spectral resolution depends on how detailed the nomenclature is.

The purpose of this study was to evaluate the capabilities of a future hyperspectral spaceborne sensor (HYPXIM) for a specific urban environmental application (urban tree classification). Given the satellite's revisiting capacity, it can be assumed that VHRS hyperspectral imaging time series would greatly improve the classification of urban tree diversity through phenological differentiation [18] while considering the development of new methods to cope with the increase size of such a data set. Recently, others innovative methods have proven their relevance for hyperspectral data and techniques: object-based or textural information are relevant for detecting vegetation types [59]; deep-learning methods and other neural networks efficiently addressed the large dimensionality of hyperspectral images for classification or dimensional reduction [60,61].

### 4.3. Towards an Urban Sampling Strategy for Mapping Urban Tree Diversity?

This study shows that the accuracy of urban tree diversity classification strongly depends on the field dataset which in turn depends on the defined nomenclature. Considering the producer's and user's accuracy of tree classification (c.f. Supplementary Table S2), it is advisable to define in advance an appropriate nomenclature based on a discriminating analysis of species and families classes. Indeed, some family classes can be better separated thanks to their spectral signature than two different species, or inversely. For instance, *Tilia tomentosa* (Silver linden) and *Tilia disticum* (Linden) species are well distinguished while belonging to the same family. Spectra variability can vary according to many factors inherited from urban morphology and urban tree management: street width and orientation, the shading effects of buildings or nearby species, tree age and size [57,58], etc.

Another limitation is the unbalanced data set that is representative of the distribution of trees generally found in urban areas. In the past, urban vegetation plans often explained why a few species dominate in cities. The diversity of trees is also due to individual initiatives. However, if a balanced

sampling strategy can be set to minimize their influence on the classification accuracy, their precise geographical identification may be difficult because private yards are not easily accessible. In this study, the Platanaceae family is represented by a single species (London plane), which is not the case for other families. This can affect the classification accuracy as some classes are over-represented, resulting in an underestimation of Silver Linden and misclassified Linden species.

However, hyperspectral data have limited classification capabilities with a limited number of training samples [62] due to possible over-adjustment to learning data [63]. Increasing the -folds in the learning process can help to minimize the influence of the unbalanced sampling by preventing from overfitting. Although further studies are needed to define an optimal sampling strategy for urban areas, we assume this strategy must be adapted to the pursued objective: Do we want to have a good map of all the species/families in a considered city? Do we want to map the dominant/specific species? While the in situ dataset should be well defined at the beginning of the study, other techniques may also be of interest to evaluate. For example, object-based image analysis (OBIA) which aim is to segment an image based not only on averaged or pixel majority spectral signatures but also on their spatial arrangement [64–67], can be helpful in minimizing neighboring misclassifications by reducing pixel-level noise while combining spectrally similar pixels.

## 5. Conclusions

This research aimed to assess the ability of VHSR hyperspectral satellite data in order to discriminate against urban tree diversity. The influence of hyperspectral techniques (dimension reduction and classification methods as well as learning data and techniques) on the classification accuracy was also evaluated on the case study of Toulouse, based on two sublevels of taxonomic nomenclature (species and families) with a minimal number of individual samples per class (4 to 10). All datasets are available upon request for the moment and will be freely available once published.

This study highlighted the great potential of VHRS (4 m) hyperspectral images (192 spectral bands). Combined with MNF dimension reduction method, the SVM classifier and a 10-folds learning method with 10 individuals per class, the discrimination of 14 species reached an overall accuracy of 78.4% (KIA = 0.73) which is consistent with the existing literature. This performance is equal to that obtained with HySpex images (2 m, 408 spectral bands). By comparing the influence of each technique, it shows that the most contributory rely upon learning methods and data. Dimension reduction methods have also a significant influence on the classification accuracy: the MNF feature extraction method is the most efficient compared to feature selection methods. The selection of vegetation indices has shown interesting potential but requires further developments to optimize their selection.

Finally, although this study was conducted using simulated spaceborne data from airborne hyperspectral images for a single, not optimal date, spaceborne VHRS hyperspectral images would help to take a major step forward by providing even more consistent time series in order to distinguish urban tree diversity through their phenology. However, urban areas have inherent limitations due to the very diverse and unbalanced composition of the urban tree canopy. Further work is needed to evaluate and define a sampling strategy adapted to urban areas.

**Supplementary Materials:** The following are available online at http://www.mdpi.com/2072-4292/11/11/1269/s1, Figure S1: Results of ANOVA for the selected Vegetation indices applied on hyperspectral datasets; Table S2: Confusion matrices for the SVM classification applied with 10-Folds method on (top) reduced image HySpex at 2 m by MNF and (down) reduced image HYPXIM 4 m by MNF; Table S3: Tables showing the Producer's and User's Accuracies (PA and UA) obtained from the MNF-reduced image datasets, with RF and SVM classifiers trained with 2-folds and 10-folds for the 14 species training dataset, on the one hand, and the 12 families on the other hand.

**Author Contributions:** Conceptualization: T.H., A.L. and C.B.; methodology: T.H. and C.B.; software: E.A.-V., C.B. and A.T.; validation: E.A.-V. and C.B.; formal analysis: C.B., E.A.-V., A.L., G.M., K.T.N.; investigation: C.B., E.A.-V., A.L., G.M., K.T.N.; resources: E.A.-V., C.B. and A.T.; data curation: C.B., E.A.-V., A.L., G.M., K.T.N., X.X.; Writing—Original Draft preparation: C.B.; Writing—Review and Editing: T.H.; supervision: T.H.; project administration: T.H.; funding acquisition: T.H.

**Funding:** This research was funded by the French National Science foundation project HYEP (Hyperspectral imagery for environmental Urban planning). Project funded by the ANR 14-CE22-0016-01.

**Acknowledgments:** Thanks to Thomas Corpetti for his technical support and help for the implementation of classifications. Authors would like to thank the two anonymous reviewers for their helpful comments.

**Conflicts of Interest:** The authors declare no conflict of interest.

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
