# Peer review of "Comparison of Hyperspectral Techniques for Urban Tree Diversity Classification"

_remotesensing, doi:10.3390/rs11111269_

Round 1
Reviewer 1 Report
Comparison of Hyperspectral techniques for improving urban tree diversity classifications
In this paper a series of techniques for the classification of trees in urban environments using multispectral and hyperspectral remote sensing images have been tested and compared. Three sensors have been considered: hypspex, hypxim and sentinel-2, with resolutions ranging from 2 m to 8 m. In the tests, four dimension reduction methods (PCAs, MNF and ANOVA), two classifiers (SVM an RF) and two learning methods (kfolds with k=2,10) are applied.
The work focuses on the performance measures of known methods, so the novelty of the work is scarce. Besides, the study is limited to very basic pixelwise classifiers, while spectral-spatial classifiers, object based, texture based or neural networks have not been considered. Texture analysis, for example, seems especially suitable for detecting vegetation types [1].
[1] Feng, Q., Liu, J. and Gong, J., 2015. UAV remote sensing for urban vegetation mapping using random forest and texture analysis. Remote sensing, 7(1), pp.1074-1094.
On the other hand, being the study a practical case, it could be useful as a reference for similar works. In that sense, it would be interesting to give more details about the problem considered, for example, tree species and their characteristics (deciduous trees, conifers, shrubs). Some of those details have been given in the discussion and in the supplementary material, but they could be completed and systematized. It would also be interesting, as far as possible, to make datasets and ground truths available (or at least the exact way to generate them), for comparisons with later works. Of course, if the manuscript is finally accepted.
Author Response
Reviewer 1
Comparison of Hyperspectral techniques for improving urban tree diversity classifications
In this paper a series of techniques for the classification of trees in urban environments using multispectral and hyperspectral remote sensing images have been tested and compared. Three sensors have been considered: hypspex, hypxim and sentinel-2, with resolutions ranging from 2 m to 8 m. In the tests, four dimension reduction methods (PCAs, MNF and ANOVA), two classifiers (SVM an RF) and two learning methods (kfolds with k=2,10) are applied.
The work focuses on the performance measures of known methods, so the novelty of the work is scarce. Besides, the study is limited to very basic pixelwise classifiers, while spectral-spatial classifiers, object based, texture based or neural networks have not been considered. Texture analysis, for example, seems especially suitable for detecting vegetation types [1].
[1] Feng, Q., Liu, J. and Gong, J., 2015. UAV remote sensing for urban vegetation mapping using random forest and texture analysis. Remote sensing, 7(1), pp.1074-1094.
Response of Authors (RoA): You are right and your comment sounds very valuable. The aim of the paper was to only focus on common-used hyperspectral techniques. While innovative methods – as those mentioned – are widely published, a comparative study on their respective influence is surprisingly not so common. As a basis, this study constitutes a first step before deepen the performance of these innovative methods and others (such as deep-learning, etc.). In any case, we added few sentences on this issue in the discussion (Prospects section) and clarify the aim of the study in the introduction.
Introduction
The aim of this study is to compare and assess which hyperspectral techniques and combination show greatest performances. Moreover, as spatial and spectral resolutions may influence classification accuracy, another objective consists in assessing which resolution is the most appropriate for urban tree diversity classification. This paper focuses only on commonly-used hyperspectral techniques as their respective influence is rarely assessed.
Discussion
Recently, others innovative methods have proven their relevance for hyperspectral data and techniques: Object-based or textural information are relevant for detecting vegetation types [xx]; deep-learning methods and other neural networks efficiently addressed the large dimensionality of hyperspectral images for classification [79, 80] or dimensional reduction [81].
[xx] Feng, Q., Liu, J. and Gong, J., 2015. UAV remote sensing for urban vegetation mapping using random forest and texture analysis. Remote sensing, 7(1), pp.1074-1094
On the other hand, being the study a practical case, it could be useful as a reference for similar works. In that sense, it would be interesting to give more details about the problem considered, for example, tree species and their characteristics (deciduous trees, conifers, shrubs). Some of those details have been given in the discussion and in the supplementary material, but they could be completed and systematized.
RoA: Accordingly to your comment and those of the 2nd reviewer, we better described the considered species in the manuscript (and not even more in the supp. mat) in the material and methods section and completely modified the Table 3 describing the species.
It would also be interesting, as far as possible, to make datasets and ground truths available (or at least the exact way to generate them), for comparisons with later works. Of course, if the manuscript is finally accepted.
RoA: The dataset is available upon request for the moment, but will be publicly available soon. We are working on publishing this dataset and upload it on a public data repository. This is mentioned in the conclusion. Indeed, publishing the dataset require further time as additional informations from all partners of the project are required for publishing it. But, they all agreed with reviewer’s comment on the interest to make images and ground truth datasets available.

Reviewer 2 Report
The article presents the comparison of hyperspectral reduction techniques in urban trees classification, taking into account families and species. The authors used in-situ data to training and validation based on 2- and 10-fold stratification. The classifiers were well known methods as Support Vector Machines and Random Forest. The datasets were HySpex and simulated HYPXIM and Sentinel-2. In most of subsection titles the authors propose hyperspectral techniques, data, remote sensing etc. but it includes Sentinel in analysis, it is some inconsistency. If Sentinel data were selected only for comparison it should be explained but the authors mentioned that sometimes for this data they reached better results so its role is rather not marginal.
The results show the accuracies obtained for different datasets, methods of sampling and classification, but they are not well ordered and sometimes it is unclear, a short subsections should be avoided.
In the tables with AOAs are some values, not so high, but further more than 70% appears. I think that more interesting could be to see such comparison and the increase of the results comparing to used datasets, methods etc. because it is essential to this work – referring to the title and “improving” word. Moreover, in supplementary materials the error matrices appears, so I think that the producer and user accuracies could be also compared with previous datasets
I suggest to check carefully the text and to be consistent with naming, e.g. Hyspex-HySpex, Hypxim-HYPXIM, Kfold-k-fold, species-Species etc.
Detailed comments I included below:
Page 1, title – is the “diversity” a good word here? Did you really classify the diversity?
Page 1, line 23 – unnecessary space
Page 1, lines 19-20 – why did you reduce the information from all data while here you stated that the influence of dimension reduction and supervised learning methods is assessed only on hyperspectral images? What is more, what is the difference between supervised learning methods and conventional classifiers (SVM and RF) ? You should explain here that it mean the method of sampling, because it could be confusing
Page 2, Table 1 – there is state of the art. So I recommend you to put more examples of such works, in recent years there are greater number of researches comparable to your study
Because of hyperspectral satellite data you could mention about some works including identification of trees or another vegetation on existing or simulated data, it could be a good start to discuss your aim where you want to assess which resolution enables to do
Page 2, line 60 – a typo (hyperspectral)
Page 2, lines 69-73 – some brief explanation of these methods
Page 3, lines 113-118 – please, add a map with the location of study area
Page 4, line 128 – please explain Cochise method and put the reference
Page 4, Table 2 – please, specify these characteristics also for Sentinel-2
Page 4, lines 136-137 – what is the basis of this assumption?
Page 5 – please add the model of equipment
Page 5, line 147, 149 – here you stated that Figure 2 sums up the main hyperspectral techniques and you tested DR methods for hyperspectral images but Figure 1 contains also Sentinel-2 which as we know is not hyperspectral.
Figure 2 – Kfolds or k-folds?
Page 6, line 172 – here you also mentioned hyperspectral data, what’s with Sentinel-2? I understand from lines 161-162 that such amount of indices was not possible to obtain but is the data included in this 14 reduced datasets or no? It is unclear
Page 6, Table 3 – I believe that if you will put the exact number for each specie it will be more valuable because when you highlight the fact that your dataset is unbalanced we don’t exactly know what is the number of individuals belonging to a particular class
Page 6, lines 184 – Kfolds or as e.g. in 189 – k-folds? Please, be consistent in whole text
Page 6, lines 184-190 – maybe it could be better explained using some graphical representation of this division
Page 6, lines 193-202 (page 7) – the description of SVM is more general than of RF, but in both cases please specify which parameters did you choose and why.
Page 7, 217 – here you stated that you calculated omission and commission errors but I cannot see this in whole manuscript. In Supplementary materials you included producer and user accuracies but not the errors
Page 7, line 229, 231, 232 – why Chlorophyll and Carotenoid starts with uppercase?
Page 7, line 233 – unnecessary space
Page 7, line 232 – Hypxim or HYPXIM?
Page 7, line 235 – I do not prefer to put such short subsection without any results presented as map, table etc.
Page 8, Table 4 – are the datasets contains Sentinel data? I don’t know how many variables contains each dataset in the table
Page 8, line 261 – as previous, maybe you could join this subsection with another, or if not, please expand it
Page 10, Figure 3 and 4 – there is no need to two numbers after dot
Page 10, line 302 – for e.g.?
Page 11, Figures – why only these data you presented on a map? I would like to see also Sentinel 2 product (referring to your sentence at Page 12, in line 337).
Page 12, line 354 – please, rephrase
Page 12, line 352 – such sentence should be supported by references. Did another authors classify similar species, families etc.? What about their user and producer accuracies?
Supplementary materials – please, check how it should be cited in the text of manuscript
Supplementary material 2 – because the topic of the article is Comparison of Hyperspectral techniques for improving classification here I would like to see this comparison, maybe not an error matrix for each technique, but only producer and user accuracies for classes – divided into techniques. It could be informative and explain how the techniques affects the accuracies, not only for selected results
Author Response
Reviewer 2
Comments and Suggestions for Authors
The article presents the comparison of hyperspectral reduction techniques in urban trees classification, taking into account families and species. The authors used in-situ data to training and validation based on 2- and 10-fold stratification. The classifiers were well known methods as Support Vector Machines and Random Forest. The datasets were HySpex and simulated HYPXIM and Sentinel-2. In most of subsection titles the authors propose hyperspectral techniques, data, remote sensing etc. but it includes Sentinel in analysis, it is some inconsistency. If Sentinel data were selected only for comparison it should be explained but the authors mentioned that sometimes for this data they reached better results so its role is rather not marginal.
Response of Authors (RoA): Thanks for this very fruitful comment. You are right and we did not succeed in clarifying the added-value of the comparison with Sentinel 2 outcomes. We modified the introduction accordingly by (1) adding few sentences / references on Sentinel2 potential use for tree species classification, and (2) better explaining this comparison in the objectives as follow:
Remote sensing applied in urban areas offers opportunities to overcome the lack of reliable and reproducible vegetation information in private and public lands at once to more accurately assess ecosystem functions [15-17]. Some authors show the value of very high spatial resolution (VHSR) with multispectral data to distinguish tree species [18-20]. [20] obtained an Overall Accuracy (OA) of 67.22% with WorldView 2 (8 bands) but the producer’s accuracy was not good enough due to this low spectral resolution. Multispectral data such as Sentinel (10 to 20m spatial resolution) show also great potential for tree species classification [21, 22]. [23] obtained an OA of 65% for 7 tree species without exploring the full potential of Sentinel2 high repetitiveness. Hyperspectral data also tends to develop for the identification of urban vegetation. While this technology is very spectrally accurate (more than 100 contiguous and narrow bands of about 10 nm each), the spatial resolution of hyperspectral satellite data remains insufficient. […]
[21] Immitzer, M.; Vuolo, F.; Atzberger, C. First Experience with Sentinel-2 Data for Crop and Tree Species Classifications in Central Europe. Remote Sens. 2016, 8, 166.
[22] Gaia Vaglio Laurin, Nicolas Pulleti, William Hawthorne, Veraldo Liesenberg, Piermaria Corona, Dario Papale, Qi Chen, Riccardo Valentini (2016) Discrimination of tropical forest types, dominant species, and mapping of functional guilds by hyperspectral and simulated multispectral Sentinel-2 data, Remote Sensing of Environment, 176:163-176
This brief state of the art shows that the accuracy of trees species classification using VHRS hyperspectral images can be influenced by DR methods, classifiers, learning methods, defined hereafter as hyperspectral techniques. In this context, evaluating capabilities of VHSR hyperspectral sensors to classify the diversity of trees species in urban area is of prime importance. The aim of this study is to compare and assess which hyperspectral techniques and combination show greatest performances. Moreover, as spatial and spectral resolutions may influence classification accuracy, another objective consists in assessing which resolution is the most appropriate for urban tree diversity classification. This paper focuses on commonly-used hyperspectral techniques as their respective influence is rarely assessed. While focusing mainly on hyperspectral data and techniques, the comparison with results obtained with Sentinel2 data may emphasize the added-value of the formers.
The results show the accuracies obtained for different datasets, methods of sampling and classification, but they are not well ordered and sometimes it is unclear, a short subsections should be avoided.
RoA: We agree. Results section is now subdivided into three main subsections which facilitate the understanding of the analyzed results:
3.1. Dimension reductions: comparison of extracted components
This subsection focuses on the results of DR methods. It does not influence the OA of classification per se, but if not shown we would not be able to evalute the respective influence of MNF and extracted VI for instance.
3.2. Evaluation of the respective influence of hyperspectral techniques and data
This subsection focuses only on the respective influence of each technique and dataset. Of course, the analysis is not very pleasant to read, switching from a technique to another. But it is summarized and efficient.
3.3. Influence of combined techniques according to images
This subsection allows to go into details of classification results. According to reviewer’s comment, we added the analysis on UA and PA in this section.
In the tables with AOAs are some values, not so high, but further more than 70% appears. I think that more interesting could be to see such comparison and the increase of the results comparing to used datasets, methods etc. because it is essential to this work – referring to the title and “improving” word.
RoA: We agree with your comments. We mainly compare the influence of each technique on classification results rather than the increase / contribution of each of them. Thus, we tried to emphasize these results by modifying the tables and highlighting the best results. However, the analysis of their respective contribution rely to the discussion and this cannot be included in the results section. As we agree with your comment, we modify the title by removing the term “improving” which is not appropriate.
Moreover, in supplementary materials the error matrices appears, so I think that the producer and user accuracies could be also compared with previous datasets
RoA: It is now incorporated within section 3.3. (see improvements made hereafter)
I suggest to check carefully the text and to be consistent with naming, e.g. Hyspex-HySpex, Hypxim-HYPXIM, Kfold-k-fold, species-Species etc.
RoA: We are sorry for these small typo mistakes. We checked carefully and modified accordingly.
Detailed comments I included below:
Page 1, title – is the “diversity” a good word here? Did you really classify the diversity?
RoA: You are right, this term can be confusing as it has several possible meanings. For instance, it does not rely the diversities commonly-used in ecology (alpha, beta, etc.). We think that this term is nevertheless appropriate, as it can refer to the greatest number of species or families. It is now better defined in the introduction (2nd §), as follow:
In this context, urban vegetation has several interests: it provides many services such as reducing the urban heat island thanks to the shade, the evapotranspiration and the photosynthesis of the trees [7–9] or improving air quality or even urban biodiversity habitats [8]. Urban vegetation has also drawbacks like allergies due to pollen, habitat for certain birds species considered as harmful by the population. However, these services and disservices depends on the plant species, their location and their structure [6, 10–12]. For instance, their capacity to mitigate the urban heat island depends on the structure and the composition of the species on the one hand, and on their respective water needs on the other hand [13, 14]. Regarding these advantages and drawbacks, mapping urban tree diversity is therefore a strong issue, where diversity is defined here by the greatest number of species or families that can be distinguished.
Page 1, line 23 – unnecessary space
RoA: Corrected.
Page 1, lines 19-20 – why did you reduce the information from all data while here you stated that the influence of dimension reduction and supervised learning methods is assessed only on hyperspectral images? What is more, what is the difference between supervised learning methods and conventional classifiers (SVM and RF) ? You should explain here that it mean the method of sampling, because it could be confusing
RoA: You are right, this section is not clear. We modified as follow. Moreover, we removed the term “supervised” which is not appropriate here.
This research aims at assessing the capabilities of VHSR hyperspectral satellite data to discriminate urban tree diversity. Four dimension reductions methods, two classifiers are tested using two learning methods applied with four in situ sample datasets. An airborne HySpex image (408 bands / 2m) has been acquired in July 2015 from which prototypal spaceborne HYPXIM hyperspectral images at 4m and 8m and multispectral Sentinel 2 image at 10m have been simulated for the purpose of this study. Comparison is made using these methods and datasets. The influence of dimension reduction methods is assessed on hyperspectral (HySpex and HYPXIM) and Sentinel 2 datasets. The influence of conventional classifiers (SVM and Random Forest) and learning methods is evaluated on all image datasets (reduced and non-reduced hyperspectral and Sentinel2 datasets). Results show that HYPXIM 4m and HySpex 2m reduced by MNF provides the greatest classification of 14 species using SVM with an overall accuracy of 78.4% (± 1.5) and a Kappa Index of Agreement of 0.7. More generally, the learning methods have a stronger influence than classifiers or even than dimensional reduction methods on urban tree diversity classification. Prototypal HYPXIM imageries appear to be a great compromise (192 spectral bands / 4m resolution) for urban vegetation applications compared to HySpex or Sentinel 2 images.
Page 2, Table 1 – there is state of the art. So I recommend you to put more examples of such works, in recent years there are greater number of researches comparable to your study
RoA: Done. Table 1 has been modified as follow, incorporating 6 additional references
Types | Sensors | Spatial Resolution | Spectral Resolution | Overall Accuracies / Number of discriminated species | Study |
Multispectral | Pleiades | 0,7 m-Panchromatic 2,4 m-Multispectral | 4 | 63,51 % for 7 species | [15] |
Ikonos | 1 m-Panchromatic 4 m-Multispectral | 4 | 56.98 % for 7 species | [16] | |
Wordview 2 | 0.46 m-Panchromatic 1.84 m-Multispectral | 8 | 62.93 % for 7 species | ||
Hyperspectral | AVIRIS | 3,5 m | 224 | 79,2 % for 29 species | [17] |
AISA | 2 m | 63 | Summer = 48 % for 7 species | [18] | |
AVIRIS | 3,5 m | 131 | 69 % for 4 evergreen species | [18] | |
MIVIS | 4 m | 102 | 75 % for 7 families | [20] | |
Hyspex | 0,4 m | 80 | 91,7 % for 13 species | [21] | |
Hyspex | 2 m | 290 | 69,3 % for 5 species | [24] | |
Rikola | 0,7 m | 64 | 63 % for 6 species | [25] | |
APEX | 2,5 m | 288 | 90% for 7 species | [26] | |
Hyperspectral + LiDAR | AVIRIS + LiDAR | 3,5 m | 224 | 83,4 % for 29 species | [17] |
AISA + LiDAR | 2 m | 63 | Summer = 57 % | [18] | |
Hyspex + LiDAR | 2 m | 290 | 83,7 % for 5 species | [24] | |
AISA + LiDAR | 1 m | 366 | 82,12 % for 4 species | [27] |
Added references:
[15] Pu, R., Landry, S., Yu, Q., Assessing the potential of multi-seasonal high resolution Pléiades satellite imagery for mapping urban tree species. International Journal of Applied Earth Observation and Geoinformation 2018, 71, 144–158.
[21] Maschler, J.; Atzberger, C.; Immitzer, M. Individual Tree Crown Segmentation and Classification of 13 Tree Species Using Airborne Hyperspectral Data. Remote Sensing 2018. 10, 1218.
[24] Shi, Y.; Skidmore, A.K.; Wang, T.; Holzwarth, S.; Heiden, U.; Pinnel, N.; Zhu, X.; Heurich, M. Tree species classification using plant functional traits from LiDAR and hyperspectral data. International Journal of Applied Earth Observation and Geoinformation 2018, 73, 207–219.
[25] Mozgeris, G.; Juodkienė, V.; Jonikavičius, D.; Straigytė, L.; Gadal, S.; Ouerghemmi, W. 2018. Ultra-Light Aircraft-Based Hyperspectral and Colour-Infrared Imaging to Identify Deciduous Tree Species in an Urban Environment. Remote Sensing 2018. 10, 1668.
[26] Dabiri, Z.; Lang, S. Comparison of Independent Component Analysis, Principal Component Analysis, and Minimum Noise Fraction Transformation for Tree Species Classification Using APEX Hyperspectral Imagery. ISPRS International Journal of Geo-Information 2018. 7, 488.
[27] Liu, H.; Wu, C. Crown-level tree species classification from AISA hyperspectral imagery using an innovative pixel-weighting approach. International Journal of Applied Earth Observation and Geoinformation 2018, 68, 298–307.
Because of hyperspectral satellite data you could mention about some works including identification of trees or another vegetation on existing or simulated data, it could be a good start to discuss your aim where you want to assess which resolution enables to do
RoA: Thanks for this helpful comment. We modified the introduction accordingly as follow:
This brief state of the art shows that the accuracy of trees species classification using VHRS hyperspectral images can be influenced by DR methods, classifiers, learning methods, defined hereafter as hyperspectral techniques. In this context, evaluating capabilities of VHSR hyperspectral sensors to classify the diversity of trees species in urban area is of prime importance. The aim of this study is to compare and assess which hyperspectral techniques and combination show greatest performances. Moreover, as spatial and spectral resolutions may influence classification accuracy, another objective consists in assessing which resolution is the most appropriate for urban tree diversity classification. Indeed, most of listed literature (table 1) used airborne hyperspectral data with a spatial resolution below 4m width which might not be achievable for spaceborne hyperspectral sensors. This study would thus contribute to the design a future spaceborne hyperspectral sensor (HYPXIM) by assessing optimal spatial resolutions for urban applications. While limited to tree diversity classification in this study, it has already been assessed for photovoltaic panels’ detection for instance [50]. This paper focuses on commonly-used hyperspectral techniques as their respective influence is rarely assessed. While focusing mainly on hyperspectral data and techniques, the comparison with results obtained with Sentinel2 data may emphasize the added-value of the formers.
[50] M. Karoui, F. Benhalouche, Y. Deville, K. Djerriri, X. Briottet, A. Le Bris. Detection and area estimation for photovoltaic panels in urban hyperspectral remote sensing data by an original NMF-based unmixing method. in Proc. of the IEEE International Geoscience and Remote Sensing Symposium (IGARSS 2018), 20-27 July 2018, Valencia, Spain, http://hyep.cnrs.fr/images/Doc/Paper_IEEE_IGARSS_2018_msk_fzb_yd_kd_xb_alb.pdf
Page 2, line 60 – a typo (hyperspectral)
RoA: Corrected.
Page 2, lines 69-73 – some brief explanation of these methods
RoA: Done as follow.
Feature extraction is defined as a set of methods for extracting information from an image [24], such as transformative methods (PCA, MNF). PCA constructs a low-dimensional representation of the data that describes as much of the variance in the data as possible. This is done by finding a linear basis of reduced dimensionality for the data, in which the amount of variance in the data is maximal [25]. However, it does not take account of signal-to-noise Ratio (SNR) unlike the minimal noise fraction (MNF) [26, 27]. MNF is also a component transformation providing ordered noise-reduced components. Feature extraction methods (PCA, MNF) are applied on the four image datasets (HySpex 2m, HYPXIM 4m and 8m, Sentinel2). Selection of the most appropriate components is made empirically since they explain more than 98% of variance.
Page 3, lines 113-118 – please, add a map with the location of study area
RoA: Done.
Page 4, line 128 – please explain Cochise method and put the reference
RoA: We added some explanations and two references:
Atmospheric corrections have been applied using the Cochise (atmospheric COrrection Code for Hyperspectral Images of remote-sensing SEnsors) method [51, 52]. This method assumes a flat ground hypothesis and estimates the water vapor content using a linear regression report (LIRR). It is well suited for hyperspectral imagery [52].
[51] Miesch C., Poutier L., Achard V., Briottet X., Lenot X., Boucher Y., (2005) Direct and Inverse Radiative Transfer Solutions for Visible and Near-Infrared Hyperspectral Imagery; IEEE TGARS, Vol. 43, N°.7, July 2005, pp 1552-1562
[52] Roussel G., Weber C., Briottet X., Ceamanos X. (2017) Comparison of two atmospheric correction methods for the classification of spaceborne urban hyperspectral data depending on the spatial resolution. International Journal of Remote Sensing, 39 (5), pp.1593 - 1614.
Page 4, Table 2 – please, specify these characteristics also for Sentinel-2
RoA: Done. Table 2 has been modified as follow
Sensor | Spatial Resolution | Bands | Dynamic Range | Spectral Resolution | |
HySpex | 2 m | 408 | 0,4-2,5 µm | 0.41 to 0.96 µm = 3.64 nm 0.96 to 2.5 µm = 6 nm | |
HYPXIM | 4 m | 192 | 10.9 nm | ||
HYPXIM | 8 m | 192 | |||
Sentinel 2 simulated | 10 m | 4 | 0,4-0,8 µm | 38-145 nm | |
6 | 0,7-2,2 µm | 18-242 nm | |||
3 | 0,4-1,3 µm | 26-75 nm | |||
Page 4, lines 136-137 – what is the basis of this assumption?
RoA: As the study area is located in the city center, there is a high probably that trees have not been cut. They are nowadays part of the climate adaptation and mitigation plan of the municipality, declared as protected contributors to cooling the urban heat island and summer heat waves. However, in case of disease, some removals may occur but, to our knowledge, they were none during this period. The paragraph has been modified as follow:
A field work has been made between May 31st and June 2nd, 2017. It is assumed that between the data acquisition (2015) and the field survey (2017), the tree composition of the city has not changed. Indeed, trees are nowadays part of the climate adaptation and mitigation plan of the municipality, declared as protected contributors to cooling urban heat islands and summer heat waves. However, in case of disease, some removals may occur but, to our knowledge, they were none during this period.
Page 5 – please add the model of equipment
RoA: Done. The equipment used is DPGS Trimble Geo7x
Page 5, line 147, 149 – here you stated that Figure 2 sums up the main hyperspectral techniques and you tested DR methods for hyperspectral images but Figure 1 contains also Sentinel-2 which as we know is not hyperspectral.
RoA: You are right. There is a small mistake in Figure 2: the arrow from ‘Sentinel2’ box should not be connected to the ‘Dimension reductions’ one. This has been corrected. We also modified the figure 1 and 2 captions as follow:
Figure 1. True color composite of hyperspectral and Sentinel 2 images. A.1 HySpex image at 2m; B.1 HYPXIM at 4m; C.1 HYPXIM at 8m; D.1 Sentinel 2 at 10m. A.2 ; B.2 ; C.2 ;D.2 : Zoom in the city center - RGB (µm) : 0.665/ 0.560/ 0.490.
Figure 2. General methodological approach. Box 1 lists the tested Dimension Reduction methods providing 14 reduced hyperspectral datasets and compared with the 3 unreduced original ones and the simulated Sentinel2 data – MNF and PCA are also applied on Sentinel2 (cf. section 2.4.2). Box 2 lists the training data and methods used to train and validate the classification (cf. section 2.4.3). Box 3 represents the two classifiers and criteria used to evaluate classifications’ accuracy (cf. section 2.4.4).
Figure 2 – Kfolds or k-folds?
RoA: This has been corrected for the whole manuscript.
Page 6, line 172 – here you also mentioned hyperspectral data, what’s with Sentinel-2? I understand from lines 161-162 that such amount of indices was not possible to obtain but is the data included in this 14 reduced datasets or no? It is unclear
RoA: You are right, it is unclear. We modified it as follow:
Finally, DR methods provide 14 reduced hyperspectral datasets which will be compared with original (unreduced) hyperspectral and Sentinel2 datasets to assess their influence.
Page 6, Table 3 – I believe that if you will put the exact number for each specie it will be more valuable because when you highlight the fact that your dataset is unbalanced we don’t exactly know what is the number of individuals belonging to a particular class
RoA: Thanks for this comment. We added two new tables accordingly describing the species / families and the number of individuals belonging the each class.
Page 6, lines 184 – Kfolds or as e.g. in 189 – k-folds? Please, be consistent in whole text
RoA: This has been corrected for the whole manuscript.
Page 6, lines 184-190 – maybe it could be better explained using some graphical representation of this division
RoA: We tried to improve the explanation as follow, but we are not sure that a graphical representation would help.
The training method is based on stratified k-folds [40]: the original dataset is randomly partitioned into k subsamples where each class is represented with the same proportion of individuals. To assess the influence of the training size, two approaches are considered. The first one uses k-folds = 2 (2-folds): the dataset is divided into 50 % of training and 50 % of validation samples and applied for the 4 training datasets. More precisely, for each class, half of individuals are used to extract the spectral signature (or assimilated values of reduced datasets) to train the classifier and the other half is used to validate the resulting classification. In order to optimize the learning capacity of classifiers, each folds is used iteratively as training and validation samples. The second one is similar to the first one and uses k-folds = 10 (10-folds). It is implemented only for training datasets with at least 10 individuals per class (“12 Families” and “14 Species”).
Page 6, lines 193-202 (page 7) – the description of SVM is more general than of RF, but in both cases please specify which parameters did you choose and why.
RoA: We detailed a little this paragraph. We did not specifically choose some parameters as we applied a hyperparameter optimization. We modified the § as follow:
Two commonly used classification methods with hyperspectral data, but not only, are tested: Support Vector Machine (SVM) and Random Forest (RF). SVM projects the original feature space into a space with a higher dimensionality to facilitate the discrimination of classes. The classifier is trained using an optimization process associated with a cost function that makes it possible to linearly separate classes to kernel functions (linear, Radial Basis Function -RBF-, Polynomial, Sigmoidal) applied on the original data allowing greater reliability and classification accuracy [29, 41]. However, it can sometimes lead to an over-fitted model with misclassification [29]. The RF classifier is an ensemble of many decision trees. This classifier is also known to be appropriate for hyperspectral data [42–44] because it tends to perform a selection of characteristics to build the decision tree during the learning process [32]. Its main advantage relies in the ability to stem over-fitting if the number of forests is sufficient. Increasing the predictive power of the model is performed here with an empirical optimization of three hyper-parameters: the number of forests, the maximum number of features, and the splitting criterion (Gini impurity or Entropy for the information gain). For both SVM and RF, hyperparameter optimization has been grid search thanks to available training datasets.
Page 7, 217 – here you stated that you calculated omission and commission errors but I cannot see this in whole manuscript. In Supplementary materials you included producer and user accuracies but not the errors
RoA: This is a mistake. We were refering to the PA and UA.
Page 7, line 229, 231, 232 – why Chlorophyll and Carotenoid starts with uppercase?
RoA: That is a mistake. Corrected.
Page 7, line 233 – unnecessary space
RoA: Corrected.
Page 7, line 232 – Hypxim or HYPXIM?
RoA: This has been corrected for the whole manuscript.
Page 7, line 235 – I do not prefer to put such short subsection without any results presented as map, table etc.
RoA: We agree. We removed all subsections titles of section 3.2. in order to be comparable with section 3.3.
Page 8, Table 4 – are the datasets contains Sentinel data? I don’t know how many variables contains each dataset in the table
RoA: Yes they are. As it is unclear, we modified the table caption as follow
Table 4. Urban tree diversity classification Averaged Overall Accuracies (AOA) and Averaged Kappa Index of Agreement (AKIA) obtained using the 2-Folds learning method and applied on each reduced hyperspectral images and on original (hyperspectral and Sentinel2) image datasets with RF and SVM according to the four sub-levels for families and species datasets.
Page 8, line 261 – as previous, maybe you could join this subsection with another, or if not, please expand it
RoA: Done.
Page 10, Figure 3 and 4 – there is no need to two numbers after dot
RoA: Corrected.
Page 10, line 302 – for e.g.?
RoA: Removed.
Page 11, Figures – why only these data you presented on a map? I would like to see also Sentinel 2 product (referring to your sentence at Page 12, in line 337).
RoA: Thanks for this comment. We were not clearly enough. We have chosen to show only classifications exhibiting the highest OA. We now mention this in the paragraphs describing these figures. Concerning the second comment, we did not show any map from Sentinel2 as the results are, in general scarce. Moreover, I think our initial statement was too optimistic because, the rare cases where S2 has greater OA than hyperspectral images (with similar techniques) concerned reduced image datasets using PCA, and for which OA do not exceed 50% (S2=47.96%, HYPXIM 8m=33.66% and HYPXIM 4m=32.74%). In some others cases, S2 has slightly higher results – for 12 families, SVM produces a classification with an OA=61.9% compared to OA=59.52% with HYPXIM8m. But these accuracies remains scarce and I am not convince by the interest of showing such map. We finally modified the sentence you mentioned (line 337) as follow:
Indeed, in some few cases, the Sentinel 2 image gives similar or even slightly better results than hyperspectral images even though the overall accuracy remains quite low.
Page 12, line 354 – please, rephrase
RoA: Rephrased as follow.
These are fairly good results given that hyperspectral data were acquired in summer, which might not be the optimal season to distinguish species in comparison to late spring or fall.
Page 12, line 352 – such sentence should be supported by references. Did another authors classify similar species, families etc.? What about their user and producer accuracies?
RoA: We were referring to the literature cited in the introduction, for which (in most of the case) only the OA is given. And unfortunally, they do not consider the same species.
This study presents results similar to those described in the literature cited in the introduction.
Supplementary materials – please, check how it should be cited in the text of manuscript
RoA: Done.
Supplementary material 2 – because the topic of the article is Comparison of Hyperspectral techniques for improving classification here I would like to see this comparison, maybe not an error matrix for each technique, but only producer and user accuracies for classes – divided into techniques. It could be informative and explain how the techniques affects the accuracies, not only for selected results
RoA: All results couldn’t have been showed as they are too numerous. We chose to show the most interesting ones. We thus added Table 8 and one paragraph describing the PA and UA of tabe 8, and one supplementary material (S3) containing 8 tables, as follow.
Related confusion matrices (Table S2) and producer’s and user’s accuracies (respectively PA and UA – Tables S3) highlight the influence of the unbalanced in situ dataset. Table 8 illustrates the PA and UA for the classifications shown in figure 5 and 6.
Table 8. PA and UA for the MNF-reduced HySpex and HYPXIM 4m hyperspectral images, classified with SVM and trained with the 14 species training dataset with 2-folds and 10-folds. These PA and UA correspond to those obtained for classifications shown in figures 5 and 6, exhibiting the greastest OA.
First, the greater is the number of individuals per tree species, the better are the PA and UA. Then, UA are in most of the case greater than the PA, except for the well represented tree species. Table S2 shows that tree species with a low PA are in most of the cases mistakenly classified as London plane. Third, the number of k-folds significantly improve accuracies for under-represented tree species, particularly for MNF-reduced HySpex images. And finally, the number of missed tree species (0% of PA or UA) is slightly greater with RF than SVM for both 14 species and 12 families nomenclatures.
S3: Tables showing the Producer’s and User’s Accuracies (PA and UA) obtained from the MNF-reduced image datasets, with RF and SVM classifiers trained with k-folds=2 and 10 for the 14 species training dataset, on the one hand, and the 12 families on the other hand.

Round 2
Reviewer 2 Report
Dear Authors,
Thank you for taking into account most of my suggestions.
Merging some subsections made the text more concise. Your explanation of presenting the best results only is enough but I am still “hungry” of knowledge how the result obtained on Sentinel-2 looks like, when you could put it as Figure 7 or supplementary material it could be perfect.
After some small changes I accept the text to publication:
Something went wrong with numbering of lines (start with 45)
Page 1, lines 63 and 65 (also Page 2, line 121) – Sentinel2 or Sentinel 2? Please, unify in the whole text
Page 2, line 112 – great !
Page 2, Table 1 – I appreciate expanding this table. Please check HySpex naming in table
Page 5 – Figure, probably 1 doesn’t have the caption and number. Please, add the source of basemap and correct HySpex naming
Page 8, Table 3 – thank you, now it is clear
Page 8, line 483 – Sigmoid